# Validation Criteria for P_ET_CO_2_ Kinetics during the Hyperventilation Provocation Test in the Diagnosis of Idiopathic Hyperventilation Syndrome

**DOI:** 10.3390/jcm11216482

**Published:** 2022-10-31

**Authors:** Nathalie Yaël Pauwen, Vitalie Faoro, Françoise Boucharessas, Thierry Colot, Alexis Guillaume, Roger Sergysels, Vincent Ninane

**Affiliations:** 1Cardio-Pulmonary Exercise Laboratory, Faculty of Motor Sciences, Université Libre de Bruxelles (ULB), 1070 Anderlecht, Belgium; 2Department of Pneumology, Centre Hospitalier Universitaire St Pierre & Brugmann, 1000 Bruxelles, Belgium; 3Department of Pneumology-Physiotherapy, Iris-Sud Hospitals, 1190 Forest, Belgium

**Keywords:** dysfunctional breathing, dysfunctional breathing syndrome, hyperventilation, hyperventilation syndrome, idiopathic hyperventilation, diagnosis, post-acute COVID syndrome

## Abstract

Background: The hyperventilation provocation test (HPTest) is a diagnostic tool for idiopathic hyperventilation syndrome (HVS), encountered in some long-COVID patients. However, interpretation of the HPTest remains unclear regarding the relevant P_ET_CO_2_ values to focus on and whether subjective symptoms should be considered. This study aimed to re-evaluate objective HPTest results for diagnosing HVS by determining accurate P_ET_CO_2_ kinetics in two groups of patients previously screened via the Nijmegen questionnaire (NQ). Methods: The kinetics of P_ET_CO_2_ during the HPTest were mathematically modeled and compared between 37 HVS patients (NQ ≥23/64) and 37 healthy controls (NQ <23/64) matched for gender, age, and body dimensions. AUC values with sensitivity and specificity were calculated, and analysis was monitored in a validation cohort of 152 routine HPTests. Results: A threshold value of a less than 12.8 mmHg increment of P_ET_CO_2_ at the 5th minute of the recovery phase of the HPTest diagnosed HVS patients with excellent sensitivity (0.92) and specificity (0.84). These results were confirmed in the validation cohort, highlighting the presence of 24% false positives/negatives when diagnosing on the basis of complaints in the NQ. Conclusions: For HVS diagnosis, we suggest considering the HPTest, which can more reliably reflect the mechanisms of CO_2_ homeostasis and the response of the respiratory center to a stimulus, regardless of the subjective onset of symptoms.

## 1. Introduction

Idiopathic hyperventilation syndrome (HVS), characterized by its polymorphic symptoms, has been given as many different names over the decades as there have been authors studying it. This syndrome is common, and it has recently been suggested that it is one of the mechanisms of persistent dyspnea in SARS-CoV-2 survivors [1,2,3].

Its diagnosis, however, remains challenging [4,5]. This is notably illustrated by the fact that between the two world wars, with the advent of psychiatry and the traumatic events that marked this period, HVS was reported to be a *“psychogenic syndrome, witnessing a disorder of the autonomic nervous system”* [6,7]. However, some researchers continued to observe HVS from a physiological perspective—including capnia—pointing out that *(1)* a decrease in alveolar carbon dioxide pressure (P_A_CO_2_) could be observed during a hyperventilation attack, [8] *(2)* a predisposition to alkalosis could be induced, [9,10] and *(3)* voluntary hyperventilation maintained for 90 s in subjects with HVS could reproduce typical symptoms recognized by the patient [11].

Attention then turned toward diagnostic tests based on voluntary hyperventilation (VH). In 1970, the first provocation test was proposed by Weimann, consisting of 5 min of VH, with a sustained breathing rate (BR) from 25–30/minute, to obtain an end-tidal fraction of CO_2_ (F_ET_CO_2_) maintained between 2 and 2.5% for at least 2 min [12]. Ultimately, Hardonk and Beumer proposed challenging pCO_2_/pH homeostasis in a hyperventilation provocation test (HPTest) that became a standard diagnostic procedure [13]. They performed this HPTest in three stages: *(1)* an adaptation phase, *(2)* a voluntary hyperventilation phase where the subjects are encouraged to increase their alveolar ventilation (VE_A_) as much as possible, and, finally, *(3)* a recovery phase. It should be noted, however, that Hardonk and Beumer observed F_ET_CO_2_ kinetics at both the 3rd and 5th minute of recovery, with more noticeable differences in subjects suffering from HVS (HVS+) at the 5th minute. As their sample offered the strongest power at the 3rd minute, they ruled that, at this point, HVS+ recovered less than two-thirds (66.7%) of baseline F_ET_CO_2_ after 3 min of recovery, without giving further details on the sensitivity and specificity.

In addition to physiological hyperventilation measurements, symptom recognition may also play a role in diagnosis [11,12,14,15]. As an example, Stam considered subjects who recognized the symptoms induced during 15 min of voluntary hyperventilation to be true positives, regardless of the depth of the arterial CO_2_ drop (P_a_CO_2_) or the duration of recovery [15]. However, in 1979, Hardonk and Beumer pointed out that not all symptoms appeared systematically in all HVS+ during VH and expressed doubts about the reliability of diagnosing HVS based on symptoms [13]. Notwithstanding these doubts, Vansteenkiste et al. argued in 1991 that the recognition of familiar symptoms in the course of the HPTest was one of the criteria for a proper diagnosis of HVS [14].

Finally, Hornsveld et al. addressed this issue in their 1996 randomized, controlled trial, pointing out that symptoms can occur in the absence of hypocapnia in both HVS+ and healthy subjects (HVS-) and confirmed that an HPTest based on symptom onset is inappropriate for HVS diagnosis [16].

In the same period, the validation of the Nijmegen questionnaire as a diagnostic tool for HVS was published, which lists a series of 16 hypocapnia-related symptoms on a 4-point Likert scale of symptom frequency [17]. The predictive properties of the Nijmegen questionnaire as a diagnostic tool were estimated by the authors to be excellent at a cut-off of ≥23/64 for positivity (Sen/Sp = 0.91/0.95-PPV/NPV = 0.94/0.92) [17]. However, 30 years later, these same authors considered that the questionnaire was more suitable for monitoring the evolution of symptoms rather than for making a diagnosis of HVS [18].

On the basis of these contradicting results, it is not surprising that the assessment of HVS is not standardized: a screening questionnaire (e.g., the Nijmegen questionnaire) is often used, with or without an HPTest, and the latter may or may not be associated with the onset of recognized symptoms [17]. This clinical practice is dependent on individual interpretation of the HPTest in a context of wide areas of uncertainty [19].

Under the assumption that there may be homeostatic physiological features to distinguish HVS+ from HVS-, we re-examined the HPTest from the perspective of the objective criterion of P_ET_CO_2_ recovery kinetics.

## 2. Methods

### 2.1. Hyperventilation Provocation Test Procedure

During the HPTest, the subject was comfortably seated in an armchair and breathed into a mouthpiece with a salivary collector and a nasal clamp. Parameters of ventilation and gas exchange were analyzed via an Oxycon Pro^TM^ (Viasys Healthcare, Conshohocken, PA, USA) with digital TripleV-Volume Sensor (Jaeger, Heinsberg, Germany). As shown in Figure 1, the HPTest began with a 3-min adaptation phase during which the patient was asked to “breathe normally”. During the second phase of the HPTest (3 min of voluntary hyperventilation), the patient was then asked to increase their tidal volume (V_T_) and to support a breath rate (BR) of at least 30/minute in order to reduce their P_ET_CO_2_ by at least 50%. During the third phase of the HPTest, the recovery phase, the patient was invited to regain ‘*normal breathing*’ for 5 min, without guidance on V_T_ or BR.

### 2.2. Training Cohort

We analyzed HPTest data from 37 subjects with complaints (with a Nijmegen questionnaire score of ≥23/64, referred to as HVS+) and 37 healthy controls without complaints (with a Nijmegen questionnaire score of <23/64, referred to as HVS-) [17]. Subjects were matched for gender, age, height, and weight. Included participants performed an HPTest on the basis of a prescription from a pneumologist or cardiologist, often following a cardio-pulmonary exercise test. All subjects (patients and controls) also had to undergo a spirometry and methacholine test, which were found to be within the expected standards. Exclusion criteria included a depressive context with psychological or psychiatric support and/or treatment with antidepressants, as well as other well identified causes of dyspnea (e.g., cardiac disease, pulmonary embolism). Patients with confirmed or suspected COVID-19 prior to the day of the HPTest were excluded from the analysis and, in order to perform the HPTest, each patient had to provide a negative PCR test.

All data were collected by the same equipment and by the same operator in the pneumology department of the CHU-St Pierre hospital (Brussels, Belgium) between September 2015 and December 2020, after the study was approved by the Ethics Committee of the CHU St Pierre hospital (B076201836758). The study was registered on ClinicalTrials.gov (NCT05100290), and the first results with 20 HVS+ and 20 matched HVS- were presented at the ERS-congress in 2019 [20].

### 2.3. Validation Cohort

In order to ascertain the results in the context of routine HPTests, we compiled an exhaustive list of 212 patients who were referred to another hospital (HIS-Molière, Brussels, Belgium) for an HPTest, between 2018 and 2021. We included all patients referred by a pneumologist or cardiologist, for whom a Nijmegen questionnaire was completed and an HPTest within the norms (3 min of adaptation–3 min of hyperventilation–5 min of recovery) was registered.

The same exclusion criteria were applied as for the training cohort (a spirometry and/or methacholine test, which were found not to be within the expected standards, depressive context with psychological or psychiatric support and/or treatment with antidepressants, any well identified causes of dyspnea, or a confirmed or suspected COVID-19 episode before the day of the HPTest). At the end of this process, 152 patients (38 Nijmegen-negative and 114 Nijmegen-positive) were included for the same analysis as the training cohort (Figure 2).

### 2.4. Statistical Analyses

#### 2.4.1. Training Cohort

The P_ET_CO_2_ parameters analyzed during the adaptation phase were minimum, maximum, average, SD, and slope-β of the linear regression. The hyperventilation and recovery phases were best described by a curvilinear model with the parameters (A, A′, a, and a′) noted from the kinetic equation TAU: TAU [P_ET_CO_2(t)_
$=A+a(1-exp^{\frac{b-t}{c}})$] (Figure 1) [21]. The maximum and minimum P_ET_CO_2_ achieved (A and A’), the fall of P_ET_CO_2_ during the VH phase (a), or the increment of P_ET_CO_2_ during the recovery phase (a′) and were compared between groups via unpaired *t*-tests or Mann–Whitney tests (for non-parametric-distributed data). For clinically relevant parameters that were found to be different between the two groups, receiver operating characteristic (ROC) curves were drawn with estimation of the 95% confidence interval of the area under the curve (AUC _[95%CI]_) and the determination of the cut-off value showing the best sensitivity (Sen) and specificity (Sen/Sp).

#### 2.4.2. Validation Cohort

In a first step, the same analysis as for the validation cohort was performed in order to check the AUC and cut-offs with the best Sen/Sp. Subsequently, the false positive (FP) and false negative (FN) rates (relative to the Nijmegen questionnaire) were compared between the validation and the training cohort, using a χ^2^ or Fisher test.

For each of the tests performed, the probability of a type I error was set at 5%. The results are presented as mean ± SD.

## 3. Results

Table 1 shows that the matching of sex, age, weight, and BMI between the HVS+ and HVS- was achieved in both cohorts (*p* > 0.05). As for the Nijmegen scores, they were found to be higher for HVS+ than for HVS- subjects in both the training and validation cohorts [35 ± 8 vs. 14 ± 7; *p* < 0.0001 and 36 ± 8 vs. 16 ± 5; *p* < 0.0001, respectively].

### 3.1. Hyperventilation Provocation Test in the Training Cohort

Table 2 provides the results for each of the three successive phases of the HP Test.

During the 3 min of the adaptation phase of the HPTest, HVS+ patients had, in comparison with the HVS-, a lower average P_ET_CO_2_ (*p* = 0.001). As illustrated in Figure 3, at the end of the adaptation phase, the instantaneous P_ET_CO_2(t3)_ levels were lower in HVS+ (29.0 ± 6.9 mmHg) than in HVS- (34.8 ± 4.6 mmHg) (*p* < 0.0001). This is supported by the slope (β) of P_ET_CO_2_ change during the adaptation phase, which was negative for HVS+ (−0.9 ± 1.7 mmHg/min), while it was positive for HVS- (0.04 ± 0.7 mmHg/min) (*p* < 0.0001), and explained by an increased average VE in HVS+ compared to HVS- (13.0 ± 5.9 vs. 9.9 ± 2.6 l/min, respectively; *p* =0.013).

During the 3 min of the voluntary hyperventilation phase, the HVS+ group paradoxically demonstrated a lower average VE than the HVS- group (46.8 ± 17.7 vs. 61.6 ± 27.2 L/min, respectively; *p* = 0.006), and finally reached, as illustrated in Figure 3, an identical minimum P_ET_CO_2(MIN)_ compared to the HVS- group (15.3 ± 3.4 vs. 14.8 ± 3.0 mmHg, respectively; *p* = 0.476).

The magnitude of the fall in P_ET_CO_2,_ (from P_ET_CO_2(t3)_ at the end of the adaptation phase to the minimum P_ET_CO_2(MIN)_ achieved during the voluntary hyperventilation phase), reflected by the parameter (a) of the TAU kinetics, showed good predictive properties with an AUC _[95%CI]_ of 0.81 [0.72–0.91], and good sensitivity at the cut-off of a fall of less than 17.6 mmHg of P_ET_CO_2_ for positivity (Sen/Sp: 0.76/0.73).

Although HVS+ and HVS- subjects started the recovery phase with comparable P_ET_CO_2(A’)_ levels (*p* = 0.278), HVS+ maintained a higher VE throughout the whole recovery phase: during the first 3 min of recovery, the average VE values for the HVS+ and HVS- subject were 19.8 ± 8.6 L/min and 16.4 ± 7.0 L/min (*p* = 0.039), respectively. Corresponding values of average VE during the 5 min of recovery amounted to 18.3 ± 8.3 L/min in HVS+ vs. 13.5 ± 5.2 L/min in HVS- (*p* = 0.002).

As a consequence, as illustrated in Figure 3, a lower P_ET_CO_2_ was observed in the HVS+ group compared to the HVS- group, both at the 3rd minute of recovery (22.5 ± 5.1 vs. 28.1 ± 4.8 mmHg, respectively; *p* < 0.0001) and at the 5th minute of recovery (23.7 ± 5.5 *vs* 30.8 ± 5.1 mmHg, respectively; *p* < 0.0001).

Specifically, at the 3rd minute of recovery, the increment of P_ET_CO_2_ (Δ P_ET_CO_2_ a’_6–9min_) showed very good predictive properties (AUC _[95%CI]_: 0.87 [0.78–0.95]) with good sensitivity and specificity (Sen/Sp: 0.81/0.81) at the cut-off of a P_ET_CO_2_ increment of less than 10.6 mmHg for positivity (Figure 4).

However, it was at the 5th minute of recovery that the increment of P_ET_CO_2_ (Δ P_ET_CO_2_ a’_6–11 min_) showed excellent predictive properties (AUC _[95%CI]_: 0.91 [0.84–0.98]) and excellent sensitivity and specificity (Sen/Sp: 0.92/0.84), at the cut-off of a P_ET_CO_2_ increment of less than 12.8 mmHg for positivity (Figure 4).

### 3.2. Results of the Hyperventilation Provocation Test in the Validation Cohort and Comparison with the Training Cohort

Table 3 provides evidence that during the 3 min of the voluntary hyperventilation phase in the validation cohort, HVS+ and HVS- still achieved the same levels of minimal P_ET_CO_2_ (P_ET_CO_2(MIN)_: 13.6 ± 2.3 vs. 13.9 ± 2.2 mmHg, respectively; *p* = 0.598) and that the fall of P_ET_CO_2_ at this point of the HPTest still showed good predictive properties at the cut-off of a fall of less than 17.7 mmHg of P_ET_CO_2_ for positivity (Sen/Sp: 0.66/0.61).

Both at the 3rd minute and the 5th minute of the recovery phase in the validation cohort, a lower P_ET_CO_2_ was observed in the HVS+ group compared to the HVS- group (*p* < 0.0001). At the 3rd minute of recovery, the increment of P_ET_CO_2_ (Δ P_ET_CO_2_ a′_6–9 min_) showed very good predictive properties (AUC _[95%CI]_: 0.80 [0.72–0.88]-Sen/Sp: 0.80/0.63), at the cut-off of a P_ET_CO_2_ increment of less than 10.6 mmHg for positivity. However, the best characteristics of the recovery were once again observed at the 5th minute of the recovery phase, with an increment of P_ET_CO_2_ (Δ P_ET_CO_2_ a′_6–11 min_) showing excellent predictive properties (AUC _[95%CI]_: 0.85 [0.78–0.92]-Sen/Sp: 0.80/0.74) at the same cut-off of a P_ET_CO_2_ increment of less than 12.8 mmHg for positivity.

As shown in Table 4, we can note that, when using the cut-off observed in the training cohort at the 3rd minute of recovery (10.56 mmHg) as the gold standard, with respect to the Nijmegen questionnaire, to determine the true and false positive (TP, FP) and negative (TN, FN) rates within the training cohort, we observed a 38% (19% FN and 19% FP) rate of misdiagnosis while using a symptomatic questionnaire. The same approach within the validation cohort demonstrated a 54% (20% FN and 34% FP) misdiagnosis rate while using a symptomatic questionnaire, although the differences between the two cohorts among both HVS+ and HVS- were not significantly different (*p* = 1.000, *p* = 0.192, respectively).

This was confirmed when we applied the same procedure using the cut-off at the 5th minute of recovery (12.79 mmHg) as the gold standard, showing a rate of 24% (8% FN and 16% FP) misdiagnosis in the training cohort and 46% (22% FN and 26% FP) in the validation cohort.

The ROC curves of the increment of P_ET_CO_2_ at the 5th minute of the recovery phase in the validation cohort is illustrated in Figure 5.

## 4. Discussion

Diagnostic criteria for HVS based on the onset of symptoms and/or P_ET_CO_2_ values or ratios during an HPTest are still a subject of debate. This study suggests that objective criteria related to the kinetics of P_ET_CO_2_ changes during the HPTest, rather than experienced symptoms and/or instantaneous values of P_ET_CO_2_ or ratios of P_ET_CO_2_, may be of value for confirmation of HVS. With respect to this, the main findings of this study were derived from the kinetics of P_ET_CO_2_ during the hyperventilation and recovery phases of the HPTest.

In both the training and validation cohorts, the most valuable diagnostic criteria were, in fact, observed at the 5th minute of the recovery phase: at the end of this period of time, the cut-off value of a P_ET_CO_2_ increment less than +12.8 mmHg detected HVS+ with excellent sensitivity and specificity in both cohorts (training cohort: Sen/Sp = 0.92/0.84; validation cohort: Sen/Sp = 0.80/0.74).

This study also provides additional data during each of the three phases of the HPTest that should be discussed in light of previous studies.

### 4.1. Adaptation Phase

During the 3 min of the adaptation phase, we observed that the differences in average P_ET_CO_2_ between the HVS+ and HVS- groups were related to an increase in average VE among HVS+ subjects compared to HVS- subjects (*p* = 0.013), corroborating previous observations made by Han et al. in 1997 and confirmed in 2013 by the meta-analysis from Grassi et al. [22,23] However, in the training cohort, the characteristics of the P_ET_CO_2_ drop, at the threshold of a drop of more than |−0.158|mmHg/min, identifies 78% of HVS+ (TP) whereas it identifies only 68% of HVS- (TN).

Vansteenkiste et al. pointed out that a fall in F_ET_CO_2_ of more than 0.25% during the 5 min of the adaptation phase would be the most predictive criterion for HVS+ (Sen/Sp = 0.57/0.83) [24]. Applying this fall rate to our training cohort’s data, however, a P_ET_CO_2_ fall of more than |−0.352|mmHg/min would retain a good specificity but a very low sensitivity (Sen/Sp: 0.54/0.73). Hardonk and Beumer also emphasized a slight drop in F_ET_CO_2_ during the short adaptation phase in their study (from 60 s to 90 s), which they attributed to stress-related hyperventilation [13]. However, they reported no difference between the HVS+ and HVS- groups, leading to the conclusion that no diagnosis can be made on the basis of resting F_ET_CO_2_ [13]. It must also be stressed that they gave no further detail on the precise moment to estimate baseline F_ET_CO_2_. Altogether, these conflicting results suggest that a correct HVS diagnosis based on the analysis of the adaptation phase seems unlikely. At least three factors may contribute to explain the discrepancies between studies. First, mouthpiece breathing can change the breathing pattern, mainly by an increase in V_T_ but also by altering inspiratory and expiratory times (Ti and Te). Second, HVS+ patients may have a heterogeneous breathing pattern, and finally, in the context of a lack of consensus on the diagnostic criteria for HVS, studies may have included patients using highly variable selection criteria [25,26,27].

It must be highlighted, however, that the adaptation phase has a major impact on P_ET_CO_2_ changes during the subsequent phases of the HPTest. For this reason, it seems imperative to standardize the adaptation phase procedure to the most frequently used duration, which is 3 min of ‘*normal breathing*’ into a mouthpiece.

### 4.2. Hyperventilation Phase

In both the training and validation cohorts, neither the minimum P_ET_CO_2(MIN)_ achieved during the voluntary hyperventilation phase, nor the P_ET_CO_2(A’)_ achieved at the end of this phase, helped to distinguish between HVS+ and HVS- groups (*p* = 0.476, *p* = 0.278, respectively), despite the fact that P_ET_CO_2(t3)_ was lower at the beginning of the hyperventilation phase in the HVS+ group than in the HVS- group (*p* < 0.0001). In fact, we noticed that, despite encouragements, all participants were struggling to decrease the P_ET_CO_2_ under a threshold of ≈15 mmHg (Figure 2, Table 2). This is in line with the suggestion from Hornsveld et al. to achieve a P_ET_CO_2_ of 15 mmHg or less during a VH of 3 min, and below the recommendation of Vansteenkiste et al. (14–17.5 mmHg), Grossman et al. (17.5 mmHg), Freeman et al. (19 mmHg), or Gardner et al. (20 mmHg), after a VH from 90 to 180 sec [14,28,29,30]. When focusing on P_ET_CO_2_ rates of fall, for example, the P_ET_CO_2_ level at the end of the voluntary hyperventilation phase (P_ET_CO_2(t6)_) compared to the P_ET_CO_2_ level at the end of the adaptation phase (P_ET_CO_2(t3)_), while we observed that the P_ET_CO_2_ rate of fall was significantly less important in HVS+ than in HVS- participants (47% vs. 58%, respectively; *p* = 0.004), these rates again refer to an instantaneous value of the adaptation phase.

In the present study, the TAU model parameters (*a*) or (*a’*) for P_ET_CO_2_ kinetics during the HPTest appeared to be the best way to differentiate between HVS+ and HVS-, giving a straightforward parameter for clinical use (Figure 1, Table 2). As reported above, the analysis of this parameter (*a*) in the training cohort indicates that a fall in P_ET_CO_2_ of less than 17.6 mmHg during the voluntary hyperventilation phase can select a positive HVS+ subject with good predictive properties (AUC_[IC95%]_ = 0.81 [0.72;0.91]; Sen/Sp = 0.76/0.73), but unfortunately, this threshold did not appear to be as relevant in the validation cohort (Sen/Sp = 0.66/0.61).

### 4.3. Recovery Phase

Finally, it is during the recovery phase that the natural response of the ventilatory system to hypocapnia is most noticeable. To that extent, no instruction about the nature of the breathing pattern should be given in contrast to the previous voluntary hyperventilation phase. Many authors report that the HVS+ patients retain excessive VE from the start of the recovery phase [13,22,30,31]. This increased VE is always due to an increase in BR and sometimes in V_T_ [21]. According to Folgering et al. and many other authors, the difficulty that HVS+ patients have in retrieving a resting VE after a VH could be the expression of an increased afterload phenomenon in the respiratory center, with an eventual hypersensitivity to CO_2_ [10,13,22,31,32]. Consistently, the excessive average VE throughout the whole recovery phase of HVS+ compared to HVS- in the training cohort leaves no doubt in our experiments, whether averaged over 3 min (*p* = 0.039) or over 5 min (*p* = 0.002) (Table 2).

However, it should be noted that the HPTest imposes a rigorous procedure throughout the test, because the increment of P_ET_CO_2_ during the recovery phase is intimately linked to the values of P_ET_CO_2_ reached during the preceding phases. In our training cohort, the P_ET_CO_2_ observed at the 3rd minute of recovery (P_ET_CO_2(t9)_) was correlated with the average P_ET_CO_2(adapt)_ of the adaptation phase (HVS+: r = 0.71; HVS-: r = 0.69; *p* < 0.0001 for both). Hardonk and Beumer also identified this close relationship between resting P_ET_CO_2_ and P_ET_CO_2_ at the 3rd minute of recovery in 100 HVS+ and 100 HVS- participants (HVS+: r = 0.62; HVS-: r = 0.70; *p* < 0.05 for both) [13]. This relationship between average resting P_ET_CO_2_ and the P_ET_CO_2_ during the recovery phase was further consolidated in our training cohort at the 5th minute of recovery (HVS+: r = 0.72; HVS-: r = 0.73; *p* < 0.0001 for both).

Irrespective of the cohort studied (training cohort or validation cohort), the delayed recovery of VE and P_ET_CO_2_, previously highlighted by numerous authors, is underlined by differences in P_ET_CO_2_ kinetics between the HVS+ and HVS- groups, most pronounced at the 3rd and 5th minutes of recovery [13,16,21,22,24,29,31,33].

With a good sample power of 100 HVS+ and 100 HVS-, Hardonk and Beumer stated that at the 3rd minute of recovery, a P_ET_CO_2(t9)_ < 66.7% of the resting P_ET_CO_2_ was the criterion for positivity, showing only 12.5% misclassification [13]. Having less power with a sample size of 44 HVS+ and 32 HVS- at the 5th minute of recovery, they did not set a 5 min criterion, although they noticed that all mean differences between groups were increased. Vansteenkiste et al. has also mentioned, at the 3rd minute of recovery, a P_ET_CO_2(t9)_ < 91% of the maximal resting P_ET_CO_2_ as a positive criterion for HVS [24]. Again, all these criteria refer to a very unstable adaptation phase or to instantaneous P_ET_CO_2_ values to estimate ratios, which is particularly challenging in a clinical context.

In this study, both cohorts best established this difference in P_ET_CO_2_ recovery at the 5th minute of the recovery phase, where an increment in the training cohort of less than 12.8 mmHg in P_ET_CO_2_ can identify HVS+ with excellent predictive properties (AUC _[95%CI]_ = 0.91 [0.84; 0.98]; Sen/Sp = 0.92/0.84), as in the validation cohort (AUC _[95%CI]_ = 0.85 [0.78; 0.92]; Sen/Sp = 0.80/0.74).

Provided one considers an increment of 12.8 mmHg of P_ET_CO_2_ at the 5th minute of recovery as the ‘gold standard’, high rates of false positives (FPs) and false negatives (FNs) in the Nijmegen questionnaire are observed in both the training cohort (16% FP–8% FN) and the validation cohort (26% FP–22% FN). Although not significant, these discrepancies in the false positive and false negative rates between the two cohorts are easily explained by the differences in the physician screening procedures that lead to an HPTest. As an example, in the training cohort, in contrast to the validation cohort, patients were referred often after a cardio-pulmonary exercise test showing an abnormal ventilatory pattern.

### 4.4. Limitations of the Study

The small number of patients enrolled could be a limitation of this study, although it showed consistent results whether the sample size was 2 × 20 subjects or 2 × 37 subjects [20].

Furthermore, since the etiology of the syndrome is still unknown and no gold standard has been established, only a set of symptoms can be used to distinguish between potential HVS+ and HVS- subjects, such as those investigated in the Nijmegen questionnaire.

As a result, the absence of a valid diagnostic reference criterion presumes the presence of false positives (FP) among HVS+ and, similarly, false negatives (FN) among controls; this remains a clear limitation of the present study.

## 5. Conclusions

In conclusion, previous criteria for the diagnosis of HVS based on the onset of symptoms and/or P_ET_CO_2_ values or ratios during an HPTest have demonstrated limitations and are still being debated. The present results suggest that the CO_2_ kinetics during a voluntary hyperventilation stress test, best reflected by falls and increases in P_ET_CO_2_ during the HPTest, provides a meaningful and convenient parameter for clinical use. The most appropriate parameter for HVS+/HVS- discrimination is obtained at the 5th minute of the recovery phase of an 11-min HPTest, at a cut-off value of a P_ET_CO_2_ increment of less than +12.8 mmHg for positivity (between the beginning and the end of the 5-min recovery phase) and with excellent predictive properties (Sen/Sp: 0.92/0.84).

## Figures and Tables

**Figure 1 jcm-11-06482-f001:**
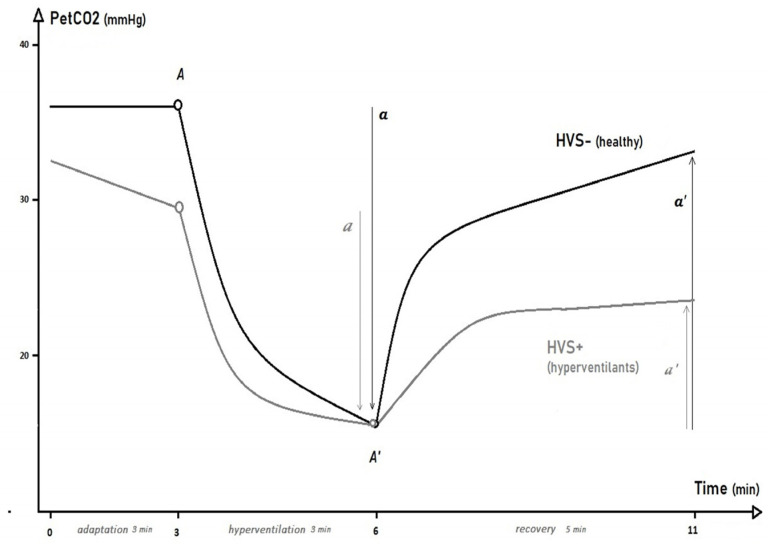
End-tidal PCO_2_ (P_ET_CO_2_) kinetics during the hyperventilation provocation test in patients with idiopathic hyperventilation syndrome (HVS+) and healthy controls (HVS−).

**Figure 2 jcm-11-06482-f002:**
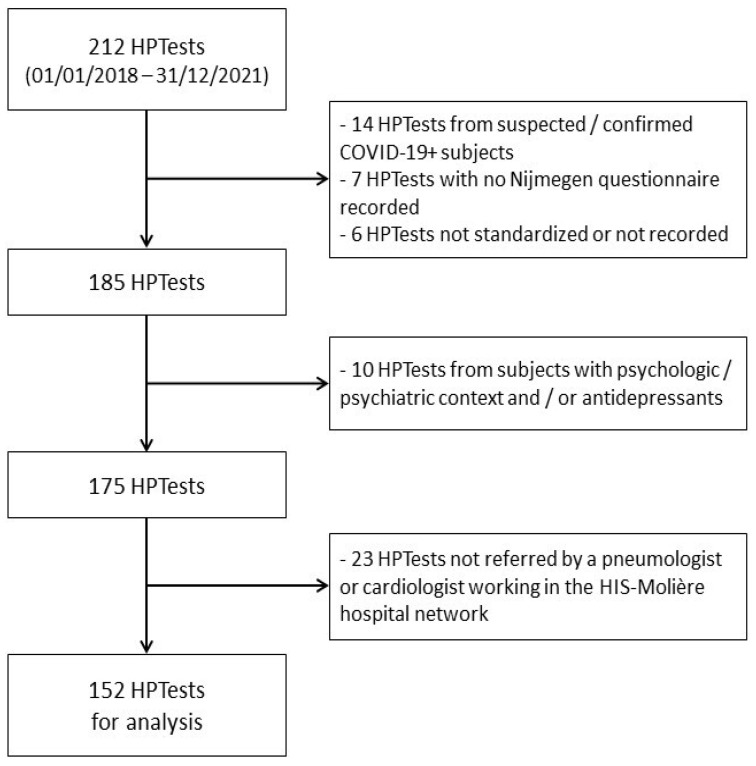
Flowchart of the inclusion/exclusion of HPTests in the validation cohort.

**Figure 3 jcm-11-06482-f003:**
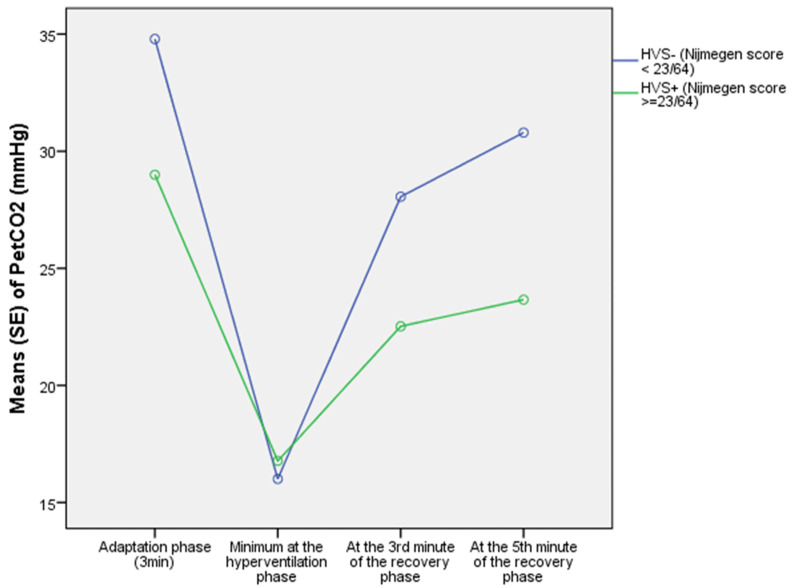
Evolution of P_ET_CO_2_ during the phases of the HPTest in HVS+ and HVS− in the training cohort.

**Figure 4 jcm-11-06482-f004:**
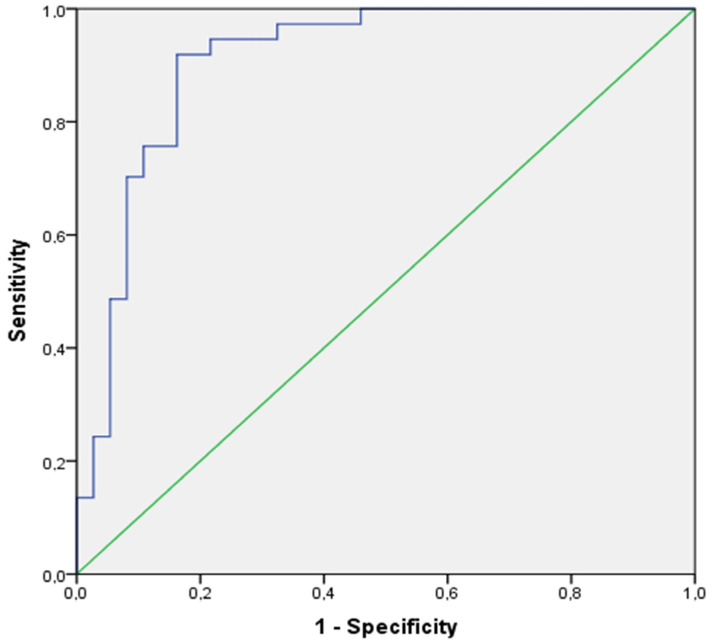
ROC curve of the increment of P_ET_CO_2_ at the 5th minute of the recovery phase in the training cohort.

**Figure 5 jcm-11-06482-f005:**
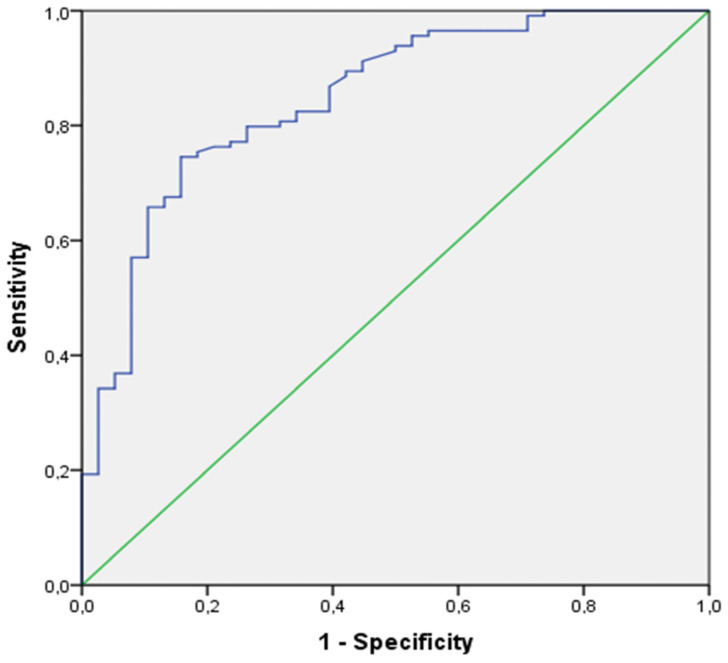
ROC curve of the increment of P_ET_CO_2_ at the 5th minute of recovery phase in the validation cohort.

**Table 1 jcm-11-06482-t001:** Baseline characteristics of the participants in the training cohort and in the validation cohort.

Training Cohort Baseline Characteristics	HVS+ (*n* = 37)	HVS- (*n* = 37)	*p*-Value
Sex *F/M n* (%)	26 (35%)/11 (15%)	26 (35%)/11 (15%)	1.0 ^†^
Age–*Yrs*	43.8 ± 13.8	44.1 ± 13.8	0.913 ^¥^
Weight–*kg*	68.6 ± 13.0	71.8 ± 14.1	0.316 ^¥^
Height–*cm*	165.5 ± 7.9	167.4 ± 7.7	0.293 ^¥^
BMI–*kg/m^2^*	25.0 ± 4.4	25.5 ± 4.6	0.615 ^¥^
Nijmegen questionnaire–*score*	34.9 ± 7.6	13.9 ± 6.8	<0.0001 ^®^
**Validation cohort** **Baseline characteristics**	**HVS+ (*n* = 114)**	**HVS- (*n* = 38)**	** *p* ** **-Value**
Sex *F/M n* (%)	72 (65.5%)/13 (12%)	17 (15.5%)/7 (%)	0.082 ^†^
Age–*Yrs*	41.3 ± 16.4	40.9 ± 21.5	0.918 ^®^
Weight–*kg*	67.9 ± 16.7	71.3 ± 18.9	0.421 ^®^
Height–*cm*	164.0 ± 8.1	167.9 ± 10.1	0.036 ^®^
BMI–*kg/m^2^*	25.2 ± 5.7	25.1 ± 5.6	1.000 ^®^
Nijmegen questionnaire–score	35.8 ± 8.4	15.6 ± 5.2	<0.0001 ^®^

^¥^ Unpaired *t*-test; ^®^ Mann–Whitney test; ^†^ Fisher test or χ^2^.

**Table 2 jcm-11-06482-t002:** Hyperventilation Provocation Test in HVS+ and in HVS- in the training cohort.

Hyperventilation Provocation Test Training Cohort	HVS+ (*n* = 37)	HVS- (*n* = 37)	*p*-Value
* **Adaptation phase** (3 min) *			
VE–*L/min*	13.0 ± 5.9	9.8 ± 2.6	0.013 ^®^
P_ET_CO_2(adapt)_–*mmHg*	31.3 ± 6.0	36.0 ± 4.3	0.001 ^®^
Slope: **β**–*mmHg/min*	−0.86 ± 1.68	+0.04 ± 0.67	<0.0001 ^®^
P_ET_CO_2(t3)_–*mmHg*	29.0 ± 6.9	34.8 ± 4.6	<0.0001 ^®^
* **Voluntary hyperventilation phase** (3 min) *			
VE _(VH)_–*L/min*	46.8 ± 17.7	61.6 ± 27.2	0.006 ^®^
P_ET_CO_2 (TAU A)_–*mmHg*	30.8 ± 6.6	36.3 ± 4.3	<0.0001 ^®^
ΔP_ET_CO_2 (TAU a)_–*mmHg*	−13.7 ± 6.3	−20.6 ± 4.7	<0.0001 ^¥^
P_ET_CO_2(VH-min)_–*mmHg*	15.3 ± 3.4	14.8 ± 3.0	0.476 ^¥^
AUC [95%CI] *of* ΔP_ET_CO_2 (TAU_–_3–6 min)_	0.812 [0.716; 0.907]		
Sen/Sp *of* ΔP_ET_CO_2 (TAU_–_3–6 min)_– (cut-off)	0.76/0.73–(−17.6 *mmHg*)		
* **Recovery phase** (3 min) *			
VE_(6–9 min)_–*L/min*	19.8 ± 8.6	16.4 ± 7.0	*0.039* ^®^
P_ET_CO_2 (A’)_–*mmHg*	15.5 ± 4.0	14.6 ± 3.2	0.278 ^¥^
ΔP_ET_CO_2 (TAU a’_–_6–9 min)_–*mmHg*	+7.3 ± 4.2	+14.9 ± 6.7	<0.0001 ^®^
P_ET_CO_2(t9)_–*mmHg*	22.5 ± 5.1	28.1 ± 4.8	<0.0001 ^¥^
AUC [95%CI] *of* ΔP_ET_CO_2 (TAU a’_–_6–9 min)_	0.866 [0.783; 0.948]		
Sen/Sp *of* ΔP_ET_CO_2 (TAU a’_–_6–9 min)_–(cut-off)	0.81/0.81–(+10.56 *mmHg*)		
* **Recovery phase** (5 min) *			
VE_(6–11 min)_–*L/min*	18.3 ± 8.3	13.5 ± 5.2	0.002 ^®^
P_ET_CO_2 (A’)_–*mmHg*	15.5 ± 4.0	14.6 ± 3.2	0.278 ^®^
ΔP_ET_CO_2 (TAU a’_–_6–11 min)_–*mmHg*	+8.3 ± 4.2	+17.8 ± 6.2	<0.0001 ^¥^
P_ET_CO_2(t11)_–*mmHg*	23.7 ± 5.5	30.8 ± 5.1	<0.0001 ^¥^
AUC [95%CI] *of* ΔP_ET_CO_2 (TAU a’_–_6–11 min)_	0.907 [0.835; 0.979]		
Sen/Sp of ΔP_ET_CO_2 (TAU a’_–_6–11 min)_–(cut-off)	0.92/0.84–(+12.79 *mmHg*)		

VE: ventilation; β: slope of linear regression of P_ET_CO_2_; P_ET_CO_2(t3)_: P_ET_CO_2_ at the 3rd minute of adaptation; VE_(VH)_: average VE during the 3 min of the VH phase; P_ET_CO_2 (TAU A)_: P_ET_CO_2_ at the start of VH phase (from TAU equation); ΔP_ET_CO_2 (TAU a)_: fall of P_ET_CO_2_ during VH phase (from TAU equation); P_ET_CO_2(VH-min)_: minimum P_ET_CO_2_ achieved during VH phase; AUC [95%CI] of ΔP_ET_CO_2 (TAU a)_: Area Under Curve of the fall of P_ET_CO_2_ during VH phase; Sen/Sp *of* ΔP_ET_CO_2 (TAU a)_: Sensitivity/Specificity of the fall of P_ET_CO_2_ during VH phase; VE_(6–9 min)_: average VE during the first 3 min of the recovery phase; P_ET_CO_2 (TAU A’)_: P_ET_CO_2_ at the start of the recovery phase (from TAU equation); ΔP_ET_CO_2 (TAU a’ –6–9 min)_: increment of P_ET_CO_2_ at the 3rd minute of the recovery phase (from TAU equation); P_ET_CO_2(t9)_: P_ET_CO_2_ at the 3rd minute of the recovery phase; Sen/Sp (a’) *of* ΔP_ET_CO_2 (TAU a’–6–9 min)–_(cut-off): Sensitivity/Specificity of the increment (a′) of P_ET_CO_2_ at the 3rd minute of the recovery phase and cut-off of P_ET_CO_2_; VE_(6–11 min)_: average VE during the 5 min of the recovery phase; P_ET_CO_2 (TAU A’)_: P_ET_CO_2_ at the start of the recovery phase (from TAU equation); ΔP_ET_CO_2 (TAU a’–6–11 min)_: increment of P_ET_CO_2_ at the 5th minute of the recovery phase (from TAU equation); P_ET_CO_2(t9)_: P_ET_CO_2_ at the 5th minute of the recovery phase; Sen/Sp (a′) *of* ΔP_ET_CO_2 (TAU a’–6–11 min)–_(cut-off): Sensitivity/Specificity of the increment (a’) of P_ET_CO_2_ at the 5th minute of the recovery phase and cut-off of P_ET_CO_2_. ^¥^ Unpaired *t*-test; ^®^ Mann–Whitney test.

**Table 3 jcm-11-06482-t003:** Hyperventilation Provocation Test in HVS+ vs. HVS- in the validation cohort.

Hyperventilation Provocation Test Validation Cohort	HVS+ (*n* = 114)	HVS- (*n* = 38)	*p*-Value
* **Adaptation phase** (3 min) *			
P_ET_CO_2(t3)_–*mmHg*	30.4 ± 5.8	33.8 ± 4.9	0.002 ^¥^
* **Voluntary hyperventilation phase** (3 min) *			
P_ET_CO_2 (TAU A)_–*mmHg*	30.7 ± 5.7	34.1 ± 4.4	0.001 ^¥^
ΔP_ET_CO_2 (TAU a)_–*mmHg*	−16.1 ± 5.3	−19.1 ± 4.5	0.002 ^¥^
P_ET_CO_2(VH-min)_–*mmHg*	13.6 ± 2.3	13.9 ± 2.2	0.598 ^®^
AUC [95%CI] of ΔP_ET_CO_2 (TAU_–_3–6 min)_	0.674 [0.579; 0.769]		
Sen/Sp of ΔP_ET_CO_2 (TAU_–_3–6 min)_–(cut-off)	0.66/0.61–(−17.7 *mmHg*)		
* **Recovery phase** (3 min) *			
P_ET_CO_2 (TAU A’)_–*mmHg*	14.8 ± 3.0	15.3 ± 3.0	0.175 ^®^
ΔP_ET_CO_2 (TAU a’_–_6–9 min)_–*mmHg*	+7.7 ± 3.7	+11.5 ± 5.7	<0.0001 ^®^
P_ET_CO_2(t9)_–*mmHg*	22.2 ± 4.4	27.1 ± 5.2	<0.0001 ^¥^
AUC [95%CI] *of* ΔP_ET_CO_2 (TAU a’_–_6–9 min)_	0.795 [0.715; 0.876]		
Sen/Sp of ΔP_ET_CO_2 (TAU a’_–_6–9 min)_–(cut-off)	0.80/0.63–(+10.61 *mmHg*)		
* **Recovery phase** (5 min) *			
P_ET_CO_2 (TAU A’)_–*mmHg*	14.8 ± 3.0	15.3 ± 3.0	0.175 ^®^
ΔP_ET_CO_2 (TAU a’_–_6–11 min)_–*mmHg*	+9.2 ± 4.4	+15.7 ± 4.5	<0.0001 ^¥^
P_ET_CO_2(t11)_–*mmHg*	23.8 ± 5.2	31.0 ± 5.1	<0.0001 ^¥^
AUC [95%CI] of ΔP_ET_CO_2 (TAU a’_–_6–11 min)_	0.847 [0.776; 0.918]		
Sen/Sp of ΔP_ET_CO_2 (TAU a’_–_6–11 min)_–(cut-off)	0.80/0.74–(+12.79 *mmHg*)		

P_ET_CO_2(t3)_: P_ET_CO_2_ at the 3rd minute of adaptation; P_ET_CO_2 (TAU A)_: P_ET_CO_2_ at the start of VH phase (from TAU equation); ΔP_ET_CO_2 (TAU a)_: fall of P_ET_CO_2_ during VH phase (from TAU equation); P_ET_CO_2(VH-min)_: minimum P_ET_CO_2_ achieved during VH phase; AUC [95%CI] of ΔP_ET_CO_2 (TAU a)_: Area Under Curve of the fall of P_ET_CO_2_ during VH phase; Sen/Sp of ΔP_ET_CO_2 (TAU a)_: Sensitivity/Specificity of the fall of P_ET_CO_2_ during VH phase; P_ET_CO_2 (TAU A’)_: P_ET_CO_2_ at the start of the recovery phase (from TAU equation); ΔP_ET_CO_2 (TAU a’–6–9min)_: increment of P_ET_CO_2_ at the 3rd minute of the recovery phase (from TAU equation); P_ET_CO_2(t9)_: P_ET_CO_2_ at the 3rd minute of the recovery phase; Sen/Sp (a′) of ΔP_ET_CO_2 (TAU a’–6–9min)_–(cut-off): Sensitivity/Specificity of the increment (a′) of P_ET_CO_2_ at the 3rd minute of the recovery phase and cut-off of P_ET_CO_2_; P_ET_CO_2 (TAU A’)_: P_ET_CO_2_ at the start of the recovery phase (from TAU equation); ΔP_ET_CO_2 (TAU a’–6–11min)_: increment of P_ET_CO_2_ at the 5th minute of the recovery phase (from TAU equation); P_ET_CO_2(t9)_: P_ET_CO_2_ at the 5th minute of the recovery phase; Sen/Sp (a′) of ΔP_ET_CO_2 (TAU a’–6–11min)_–(cut-off): Sensitivity/Specificity of the increment (a’) of P_ET_CO_2_ at the 5th minute of the recovery phase and cut-off of P_ET_CO_2_. ^¥^ Unpaired *t*-test; ^®^ Mann–Whitney test.

**Table 4 jcm-11-06482-t004:** Comparisons of the predictive properties between the training and the validation cohorts.

Sensitivity and Specificity of the HP Test in the Two Cohorts	Training Cohort (*n* = 74)	Validation Cohort (*n* = 152)	*p*-Value
* **Recovery phase** (3 min)–cut-off <10.56 mmHg *			
	**HVS+ (*n* = 37)**	**HVS+ (*n* = 114)**	
TP-True positives–*number (%)*	30 (81%)	91 (80%)	1.000 ^®^
FN-False negatives–*number (%)*	7 (19%)	23 (20%)	
	**HVS- (*n* = 37)**	**HVS- (*n* = 38)**	
TN-True negatives–*number (%)*	30 (81%)	25 (66%)	0.192 ^®^
FP-False positives–*number (%)*	7 (19%)	13 (34%)	
* **Recovery phase** (5 min)–cut-off <12.79 mmHg *			
	**HVS+ (*n* = 37)**	**HVS+ (*n* = 114)**	
TP-True positives–*number (%)*	34 (92%)	89 (78%)	0.087 ^®^
FN-False negatives–*number (%)*	3 (8%)	25 (22%)	
	**HVS- (*n* = 37)**	**HVS- (*n* = 38)**	
TN-True negatives–*number (%)*	31 (84%)	28 (74%)	0.399 ^®^
FP-False positives–*number (%)*	6 (16%)	10 (26%)	

Comparison of the sensitivity and specificity between the two cohorts (at the cut-offs observed in the training cohort at the 3rd and 5th minute of recovery) ^®^ χ^2^ test.

## Data Availability

To protect patient privacy and comply with relevant regulations, identified data are unavailable. Requests for de-identified data will be granted to qualified researchers with appropriate ethics board approvals and relevant data-use agreements.

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
