# Peer review of "Validation Criteria for PETCO2 Kinetics during the Hyperventilation Provocation Test in the Diagnosis of Idiopathic Hyperventilation Syndrome"

_jcm, 2022, doi:10.3390/jcm11216482_

Round 1
Reviewer 1 Report
Original study-Objective criteria during the hyperventilation provocation test
for the diagnosis of idiopathic hyperventilation syndrome
Thanks for the opportunity to review this original article by dr Nathalie Yaël Pauwen et al. titled:Objective criteria during the hyperventilation provocation test for the diagnosis of idiopathic hyperventilation syndrome
The study is interesting and focuses on the Idiopathic hyperventilation syndrome and the authors aim to validate an objective method to effectively diagnose it. I suggest few comments that I would like to be better addressed in the manuscript
1. Title: the title does not very fit the study, it is more a: validation criteria of PETCO2 kinetic for hyperventilation provocation test in the diagnosis of idiopathic hyperventilation syndrome. please revise it
2. Abstract: the aim was really to determine accurate PETCO2 kinetic to diagnose HVS, re-evaluating objective HPT test results among two group of patients previously screened for HVS via the Nijmegen-questionnaire. Please revise
3. Intro: please revise in a more accurate manner the last paragraph to let the reading better understand the aim of the study.( see the abstract suggestion )
4. Methods: pit is not clear in text, table and flowchart both in the methods section and in the results how many people that participate in the study already had COVID 19 and what kind of infection if mild of moderate to severe detailing the presence in both groups considered. it is unclear in particular if people with current at the time of the test or past covid infection were excluded
If there was not any people who have had covid or long covid then I would delete the sentence in the intro related to COVID19 because is misleading and not clearly related to the present study which instead is a physiology study.
Also, before using it, please describe in detail what is the Nijmegen questionnaire score and why it is used for HVS+.
Please explain better in the methods the 2 different recovery phase and why you need to distinguish among the two phases of 3 and 5 minutes ( only one recovery phases described ) that in the results are described and graphed in the table
6. Limitation: other limitations that may be mentioned are the small number of patients enrolled and the fact that the presence or absence of HVS is based on a questionnaire quality test ad not on objective test which is the scope of the present study.
There is a number of misspelling errors and some phrases that need to be rephrased in the manuscript, please provide a mother tongue English revision and mention it in the acknowledgments.
Thanks again for this opportunity. I look forward to revise the next version of the manuscript.
Author Response
Dear reviewer,
We are grateful for your careful reading and your insightful comments. We will take them up below one by one and answer them in detail.
1. Title: the title does not very fit the study, it is more a: validation criteria of PETCO2 kinetic for hyperventilation provocation test in the diagnosis of idiopathic hyperventilation syndrome. please revise it
Thank you for this point, we have modified the title in this sense :
« Validation criteria for PETCO2 kinetics during the hyperventilation provocation test in the diagnosis of idiopathic hyperventilation syndrome "
2. Abstract: the aim was really to determine accurate PETCO2 kinetic to diagnose HVS, re-evaluating objective HPT test results among two group of patients previously screened for HVS via the Nijmegen-questionnaire. Please revise
Indeed, we have also modified the objective of the study in this sense :
« This study aimed to re-evaluate objective HPTest results for diagnosing HVS by determining accurate PETCO2 kinetics in two groups of patients previously screened via the Nijmegen questionnaire (NQ) ».
3. Intro: please revise in a more accurate manner the last paragraph to let the reading better understand the aim of the study.( see the abstract suggestion )
Indeed, we have also changed the phrasing in this paragraph :
« Under the assumption that there may be homeostatic physiological features to distinguish HVS+ from HVS-, we re-examined the HPTest from the perspective of the objective criterion of PETCO2 recovery kinetics. »
4. Methods: pit is not clear in text, table and flowchart both in the methods section and in the results how many people that participate in the study already had COVID 19 and what kind of infection if mild of moderate to severe detailing the presence in both groups considered. it is unclear in particular if people with current at the time of the test or past covid infection were excluded. If there was not any people who have had covid or long covid then I would delete the sentence in the intro related to COVID19 because is misleading and not clearly related to the present study which instead is a physiology study.
Your valuable comment made us specify in the methodology the exclusion criteria for patients with suspected/confirmed Covid-19 prior to the HPTest :
« Patients with confirmed or suspected COVID-19 prior to the day of the HPTest were excluded from the analysis and, in order to perform the HPTest, each patient had to provide a negative PCR test. ”
The flow chart mentions the number (14) of patients excluded for reporting a history of Covid-19 (suspected or confirmed by a PCR+ test). We chose to exclude them in order not to confuse the purpose of the study, which focuses on HVS. For this reason no distinction was made on the type of Covid-19 infection (mild, moderate or severe). It should be taken into account that the study on HVS having started before the pandemic and the paper being published after it, taking into account the HVS observed in certain long Covid-19, it seemed useful to us to specify that we voluntarily excluded these subjects, in order to remain focused on the initial topic of the study (HVS) and not on a HVS possibly associated with long Covid-19.
5. Also, before using it, please describe in detail what is the Nijmegen questionnaire score and why it is used for HVS+.
Thank you, indeed, this had been specified in a previous version and removed. We have reported this information on the Nijmegen questionnaire in the introduction paragraph :
« In the same period, the validation of the Nijmegen questionnaire as a diagnostic tool for HVS was published, which lists a series of 16 hypocapnia-related symptoms on a 4-point Likert scale of symptom frequency. 17 The predictive properties of the Nijmegen questionnaire as a diagnostic tool were estimated by the authors to be excellent at a cut-off of ≥ 23/64 for positivity (Sen/Sp= 0.91/0.95 - PPV/NPV=0.94/0.92). 17 However, these same authors considered 30 years later that the questionnaire was more suitable for monitoring the evolution of symptoms than for making a diagnosis of HVS.18 "
6. Please explain better in the methods the 2 different recovery phase and why you need to distinguish among the two phases of 3 and 5 minutes ( only one recovery phases described ) that in the results are described and graphed in the table
Indeed, the HPTest can be concluded after the 3rd minute of recovery (as Hardonk & Beumer did) or at the 5th minute of recovery. Hardonk & Beumer also did this, observing increased differences between HVS+ and HVS- subjects at the 5th minute. Their choice to rule at the 3rd minute was only for reasons related to the size of their sample (power), which was larger at the 3rd minute. We observed both time points in our study and found that the 5th minute had the best predictive properties. Thank to your comment, this has been further clarified in the new version of the paper :
« It should be noted, however, that Hardonk & Beumer observed FETCO2 kinetics at both the 3rd and 5th minute of recovery, with more noticeable differences in subjects suffering from HVS (HVS+) at the 5th minute. As their sample offered the strongest power at the 3rd minute, they ruled at this point that HVS+ recovered less than 2/3 (66.7%) of baseline FETCO2 after 3 minutes of recovery, without giving further details on the sensitivity and specificity. »
7. Limitation: other limitations that may be mentioned are the small number of patients enrolled and the fact that the presence or absence of HVS is based on a questionnaire quality test ad not on objective test which is the scope of the present study.
Indeed, we have added these points to the limitations of the study :
« The small number of patients enrolled could be a limitation of this study, although it showed consistent results whether the sample size was 2x20 subjects or 2x37 subjects.20
Furthermore, since the etiology of the syndrome is still unknown and no gold standard has been established, only a set of symptoms can be used to distinguish between potential HVS+ and HVS- subjects, such as those investigated in the Nijmegen questionnaire.
As a result, the absence of a valid diagnostic reference criterion presumes the presence of false positives (FP) among HVS+ and, similarly, false negatives (FN) among controls; this remains a clear limitation of the present study. ”
8. There is a number of misspelling errors and some phrases that need to be rephrased in the manuscript, please provide a mother tongue English revision and mention it in the acknowledgments.
Indeed, after having been redesigned many times, the text has been revised by a native English speaker who is thanked in the ad-hoc section.
Thanks again for this opportunity. I look forward to revise the next version of the manuscript.
I thank you again for your time and the relevance of your comments.
Nathalie Pauwen

Reviewer 2 Report
Thank you for the opportunity to review your study. This is a very well-written study and the methods and results are explained nicely and in great detail. I liked the tables and figures. I suggest you elaborate on/explain the study's limitations in more detail.
Author Response
Dear reviewer,
We are grateful for your careful reading and your encouraging comments.
We have indeed developed the limitations of the study somewhat, which you will find in the ‘limitation of the study’ section :
« The small number of patients enrolled could be a limitation of this study, although it showed consistent results whether the sample size was 2x20 subjects or 2x37 subjects.20
Furthermore, since the etiology of the syndrome is still unknown and no gold standard has been established, only a set of symptoms can be used to distinguish between potential HVS+ and HVS- subjects, such as those investigated in the Nijmegen questionnaire.
As a result, the absence of a valid diagnostic reference criterion presumes the presence of false positives (FP) among HVS+ and, similarly, false negatives (FN) among controls; this remains a clear limitation of the present study. "
I thank you again for your time and the relevance of your comments.
Nathalie Pauwen

Round 2
Reviewer 1 Report
thanks again for the opportunity to revise this manuscript. the authos have addressed all the comments.
I have no further comments.